# Mental Health Problems among COVID-19 Frontline Healthcare Workers and the Other Country-Level Epidemics: The Case of Mexico

**DOI:** 10.3390/ijerph19010421

**Published:** 2021-12-31

**Authors:** Rebeca Robles, Silvia Morales-Chainé, Alejandro Bosch, Claudia Astudillo-García, Miriam Feria, Sara Infante, Natasha Alcocer-Castillejos, Leticia Ascencio, Janet Real-Ramírez, Dulce Díaz, Héctor Francisco Gómez-Estrada, Claudia Becerra, Raúl Escamilla, Alejandra López-Montoya, Ana Beristain-Aguirre, Hamid Vega, Dení Álvarez-Icaza, Evelyn Rodríguez, Sol Durand, Ana Fresán, María-Elena Medina-Mora, Carmen Fernández-Cáceres, Eduardo Ángel Madrigal de León

**Affiliations:** 1Ramón de la Fuente Muñiz National Institute of Psychiatry, Ciudad de México 14370, Mexico; mferia35@imp.edu.mx (M.F.); infante1999@yahoo.com (S.I.); psc.dulce.diaz@gmail.com (D.D.); claudiapalars@gmail.com (C.B.); recam1@hotmail.com (R.E.); hamid.vega@gmail.com (H.V.); dra.durand@imp.edu.mx (S.D.); a_fresan@yahoo.com.mx (A.F.); metmmora@gmail.com (M.-E.M.-M.); eduardo.madrigal@imp.edu.mx (E.Á.M.d.L.); 2Faculty of Psychology and General Directorate of Academic Personnel Affairs, National Autonomous University of Mexico, Ciudad de México 04510, Mexico; smchaine@gmail.com (S.M.-C.); alejandro.bosch@dgaco.unam.mx (A.B.); alelopez.unam@gmail.com (A.L.-M.); anab@dgaco.unam.mx (A.B.-A.); dradenialvarezi@gmail.com (D.Á.-I.); 3Psychiatric Care Services, Ministry of Health, Ciudad de México 11470, Mexico; claudiaiveth.astudillo@gmail.com; 4Salvador Zubirán National Institute of Medical Sciences and Nutrition, Ciudad de México 14080, Mexico; natdian@gmail.com; 5Palliative Care Service, National Cancer Institute, Ciudad de México 14080, Mexico; leash71.lah@gmail.com; 6Population Health Research Center, National Institute of Public Health, Ciudad de México 14080, Mexico; janet.real@insp.mx; 7Centros de Integración Juvenil, Ciudad de México 03600, Mexico; dirtrat@cij.gob.mx (H.F.G.-E.); cfernandez@cij.gob.mx (C.F.-C.); 8Infectious Disease Research Center, National Institute of Respiratory Diseases, Ciudad de México 14080, Mexico; evelynre13@gmail.com

**Keywords:** COVID-19, healthcare workers, mental disorders

## Abstract

COVID-19 frontline healthcare workers (FHCW) are struggling to cope with challenges that threaten their wellbeing. We examine the frequency and predictors of the most frequent mental health problems (MHP) among FHCW during the first COVID-19 peak in Mexico, one of the most severely affected countries in terms of FHCW’s COVID-19 mortality. A cross-sectional survey was conducted between May 8 and August 18, 2020. A total of 47.5% of the sample (*n* = 2218) were FHCW. The most frequent MHP were insomnia, depression, posttraumatic stress symptoms, and health anxiety/somatization (whole sample: 45.7, 37.4, 33.9, and 21.3%; FHCW: 52.4, 43.4, 40.3 and 26.1, respectively). As compared to during the initial COVID-19 phase, depression and health anxiety/somatization symptoms as well as experiences of grieving due to COVID-19, personal COVID-19 status, and having relatives and close friends with COVID-19 were more frequent during the COVID-19 peak. Obesity, domestic violence, personal COVID-19 status, and grieving because of COVID-19 were included in regression models for main FHCW’s MHP during the COVID-19 peak. In conclusion, measures to decrease other country-level epidemics contributing to the likelihood of COVID-19 complications (obesity) and MHP (domestic violence) as well as FHCW´s probability of COVID-19 infection could safeguard not only their physical but also mental health.

## 1. Introduction

Worldwide, COVID-19 frontline healthcare workers (FHCW), including medical staff and affiliated healthcare workers, are struggling to cope with challenges that threaten their wellbeing, counting a higher likelihood of contagion because of their continued exposure to patients with COVID-19. Mexico is among the most severely affected countries in terms of COVID-19 mortality, partly due to complications related to other nationwide epidemics such as obesity and diabetes [1]. According to the Ministry of Health of Mexico (https://www.gob.mx/salud/documentos/datos-abiertos, accessed on 26 April 2021), at least a third part (33.46%) of the 235,343 FHCW with confirmed COVID-19 diagnosis had a chronic illness; obesity was the most frequent (15.28%). This could exacerbate the fear of COVID-19 contagion and produce higher levels of psychological distress in those infected.

Consistently, since the initial COVID-19 phase or a scenario in which there are clusters of cases (defined by the WHO as the period when a country experiences cases limited to well-defined clusters related by time, geographic location and common exposures) [2], a large proportion of Mexican FHCW has reported clinically significant symptoms of health anxiety/somatization, with healthcare workers’ personal COVID-19 status constituting the main risk factor for presenting insomnia and posttraumatic stress symptoms (PTS) [3]—above well-known individual risk factors for affective and stress-related disorders, such as being a woman [4,5]. Moreover, COVID-19 status among healthcare workers’ relatives or people close to them could even double their risk of depression and PTS, while mourning the death of loved ones or people close to them because of COVID-19 was their main risk factor for depression [3].

Social and domestic violence is another national problem with an extremely negative impact on mental health [6]. Mexico is one of the countries where verbal and even physical aggression has been directed against COVID-19 FHCW [7], and FHCW are not exempt from the country’s high domestic violence rate. Since the initial COVID-19 phase or the clusters-of-cases scenario in Mexico, the presence of domestic violence has doubled the risk for the most frequent mental health problems (MHP) in healthcare workers [3].

Unfortunately, all these experiences of personal contagion and that of people close to them, and grieving over the death of loved ones were expected to increase during the COVID-19 community transmission scenario (when outbreaks occur with the inability to relate confirmed cases through chains of transmission for a large number of cases) [2], thereby raising the risk of MHP among FHCW. In this scenario, monitoring FHCW’s mental health status and associated risk factors during the most stressful stage (the peak of the community transmission scenario) is a priority for informing cost-effective strategies to prevent long-lasting negative effects of chronic emotional distress on FHCW’s health and wellbeing, to enable their recovery from this traumatic event, and be better prepared for possible future COVID-19 outbreaks and health emergencies in Mexico and other similar countries.

In this vein, the aim of this study was to determine the frequency of MHP and related personal, social, and COVID-19-related risk factors among FHCW during the first peak of the COVID-19 community transmission scenario in Mexico. Specific objectives included: (1) comparing the frequency of MHP between FHCW and healthcare workers not working directly with COVID-19 patients; (2) comparing FHCW’s frequency of MHP between the COVID-19 clusters-of-cases scenario and the peak of COVID-19 community transmission scenario; and (3) determining the level of risk conferred by personal, social, andCOVID-19-related variables on the main MHP reported by FHCW during the peak of the COVID-19 community transmission scenario in Mexico.

We expected that, as was observed during the initial COVID-19 phase in Mexico [3], insomnia, depression, PTS, and health anxiety/somatization symptoms would be the most frequent MHP among healthcare workers, and that higher frequencies would be reported by those working directly with COVID-19 patients (FHCW). In line with the anticipated increase in MHP prevalence among FHCW dealing with COVID-19 [8,9], we hypothesized that frequencies of MHP among FHCW during the community transmission scenario would be higher than those reported by FHCW during the clusters-of-cases scenario, particularly those related to the increase in grieving over the death of loved ones due to COVID-19 and personal COVID-19 status (i.e., depression and health anxiety, respectively) [10,11].

## 2. Materials and Methods

A national, online system for the identification and treatment of MHP among HCWs was set up by the Ministry of Health in collaboration with the National Autonomous University of Mexico (UNAM), which included a consent form and evaluation tool on the official COVID-19 website (coronavirus.gob.mx/salud-mental/) programmed on a platform (https://www.misalud.unam.mx/covid19/, accessed on 2 December 2021) developed in Linux (Linux Kernel Organization, Inc., Beaverton, OR, USA)^®^, PHP (PHP Systems/Design, Houston, TX, USA)^®^, HTML (World Wide Web Consortium (W3C), Cambridge, MA, USA)^®^, CSS (CSS Working Group, Harlow, UK)^®^, and JavaScript (Netscape Communications, Mozilla Foundation, San Francisco, CA, USA)^®^, which encrypts the information under the computer security standards of the Mexican laws on protection of personal data. This allowed us to ask for participants’ Unique Population Registry Code when loging into the system to ensure traceability of participation although it was anonymized.

Mexican HCWs were invited to participate in the survey through official media (such as the federal government’s microsite—coronavirus.gob.mx/salud-mental/, social networks (including Facebook and Twitter), and press conferences, and the National Institute of Psychiatry’s website and social networks). During the survey, participants must answer all items to be allowed to proceed to the next question (to avoid missing data). At the end of the survey, participants received feedback on their mental health status, preventive measures for individuals with a low risk of mental health problems (including videos and online courses), and specific contact information (telephone and cell phone numbers and e-mails) for those requiring specialized mental health treatment.

A convenience sample of Mexican nurses, general practitioners, medical residents and medical specialists was integrated by inviting healthcare workers dealing with COVID-19 to complete an online cross-sectional survey from 8 May (when Mexican Undersecretary of Health declared the second phase of the pandemic, describing it as the community spread stage in Mexico) to 18 August 2020 (when according to the Mexican Undersecretary of Health, the country had reached the “catastrophic scenario” -exceeding 60 thousand deaths due to COVID-19- and started with a trend of case reduction for two weeks). The study target population includes, approximately 315,000 nurses [12], 270,600 general practitioners [13], 22,613 medical residents [14], and 147,910 medical specialists [15].

All study procedures and materials, including consent forms, were previously approved by the Institutional Review Board (IRB) of the Ramón de la Fuente Muñiz National Institute of Psychiatry, Mexico (ref: CEI/C/010/2020). Informed consent was obtained from all participants. The survey met the criteria for internet E-surveys such as informed consent, data protection, development, testing, contact mode, advertising the survey, mandatory, voluntary completion, cookies used, IP check, log file analysis, registration, and atypical timestamp considerations [16].

### 2.1. Variables and Measures

The evaluation comprised two general sections. The first was a set of self-reported questions on personal, social, and COVID-19-related variables (gender, age, marital status, education, location, occupation, obesity, domestic violence, personal COVID-19 status, COVID-19 status of relatives or close persons, being COVID-19 FHCW, and grieving for the death of loved/close ones due to COVID-19.

The second section included a compilation of various scales and questionnaires: (1) The PTSD checklist for DSM-5 (PCL-5) [17] according to DSM-5 diagnostic criteria [18], as well as insomnia (through the item “Trouble falling or staying asleep?”); (2) The 5-item Anxiety Scale for ICD-11 PHC field studies, in which a total score of three or more predicted 89.6% of above-threshold cases with generalized anxiety [19]; (3) The first eight items in the SSOM Current Status Assessment Questionnaire, a self-report measure used to identify somatoform symptoms and disorders [20]; and (4) The Patient Health Questionnaire-2 (PHQ-2), an accurate, ultra-brief screening tool for depression [21,22], in which the recommended cut-off point is a score of three or over [23]. Specific questions for this study are available in a pdf format at: https://1drv.ms/u/s!AmLeJVltqurSm0RtdVhZhp__ARhO?e=gA58Xc, accessed on 2 December 2021).

### 2.2. Procedures and Data Analyses

A national, online system for the identification and treatment of MHP among HCWs was set up by the Ministry of Health in collaboration with the National Autonomous University of Mexico (UNAM), which included the consent form and evaluation tool described earlier on the official COVID-19 website (coronavirus.gob.mx/salud-mental/, accessed on 2 December 2021). Mexican HCWs were invited to participate in the survey through official media (such as the federal government’s microsite—coronavirus.gob.mx/salud-mental/, accessed on 2 December 2021, social networks (including Facebook and Twitter), and press conferences, and the National Institute of Psychiatry’s website and social networks). At the end of the survey, participants received feedback on their mental health status, preventive measures for individuals with a low risk of mental health problems (including videos and online courses), and specific contact information (telephone and cell phone numbers and e-mails) for those requiring specialized mental health treatment.

All analyses were performed using SPSS Statistics 21.0. First, we obtained descriptive statistics to characterize the sample, using means and standard deviations to describe continuous variables and frequencies and percentages to summarize categorical ones. Cronbach’s alpha coefficients were used to analyze the internal consistency of the measures in our sample. Independent samples, Student-t tests, and chi square tests were used to compare these variables between the present sample of healthcare workers during the first peak of the community transmission scenario in Mexico, and Robles et al.’s [3] sample of healthcare workers during the clusters-of-cases scenario.

Next, Chi square tests were calculated to compare the frequency of MHP between the following (a) FHCW and healthcare workers not working directly with COVID-19 patients during the peak of the community transmission scenario in Mexico (present sample), and (b) FHCW in the clusters-of-cases scenario sample [3] and the first peak of community transmission scenario in Mexico (present sample).

Finally, we analyzed the relationship between the most frequent MHP in FHCW during the COVID-19 community transmission scenario in Mexico (including insomnia, depression, PTSD, and health anxiety/somatization) with personal, social, andCOVID-19-related variables evaluated in the study, given their potential effect on mental health status among FHCW. Initially, Chi-square tests were used for bivariate analyses to identify possible predictors for each MHP. Variables that achieved *p* ≤ 0.01 in the bivariate analyses were included in a subsequent multivariable logistic regression analysis for each MHP. A backward elimination process was employed to identify the most explanatory calibrated regression model and the Hosmer-Lemeshow test was used to determine the model’s goodness of fit. Results including an associated *p*-value of at least ≤0.05 were deemed statistically significant. The most calibrated model is presented as the final regression model for each MHP. Odds ratios (ORs) with a 95% confidence interval (95% CI) and *p*-values are reported.

## 3. Results

Table 1 presents the description of personal, social, and COVID-19-related variables in the total sample of this study (*n* = 4670 healthcare workers during the COVID-19 community transmission scenario in Mexico) in comparison with the characteristics of the sample of healthcare workers dealing with COVID-19 during the clusters-of-cases scenario in Mexico (*n* = 5938) [3]. As can be seen, previously reported risk factors for MHP, such as gender, marital status, and domestic violence, remained unchanged, while the frequencies of personal COVID-19 status (or suspected/confirmed COVID-19 diagnosis) and grieving for the death of people close to them due to COVID-19 more than doubled during the community transmission scenario.

Table 2 provides a comparison of MHP frequencies between FHCW (47.5%, *n* = 2218) and healthcare workers not working directly with COVID-19 patients (52.5%, *n* = 2452) during the peak of the community transmission scenario in Mexico. In general terms, all MHP were more frequent among FHCW. Insomnia, depression, PTS, and health anxiety/somatization were the main MHP in both frontline and non-frontline healthcare workers. When compared with FHCW dealing with COVID-19 during the clusters-of-cases scenario (also in Table 2), depression and health anxiety/somatization were more frequently reported among FHCW during the first peak of the community transmission scenario. According to our analyses on the internal consistency of the measures employed to evaluate the presence of these mental disorders showed high Cronbach alphas in our sample, being 0.939 for PCL-5, 0.849 for the Anxiety Scale for ICD-11 PHC field studies, 0.897 for the first eight items in the SSOM Current Status Assessment Questionnaire, and 0.874 for PCL-2.

To further explore the association between these variables and the most frequent MHP, specifically among the subsample of FHCW during the first peak of the COVID-19 community transmission scenario (*n* = 2218), Table 3 presents significant bivariate comparisons by MHP. Obesity, domestic violence, personal COVID-19 status, and grieving for the death of close ones due to COVID-19, were more frequent in FHCW with any MHP (including depression and health anxiety/somatization) than among those without this diagnosis. Additionally, being a woman made FHCW more prone to all main MHP, and also being single, ≤39 years old and/or graduate to insomnia, depression, and PTS.

Moreover, when these personal, social, and COVID-19-related variables were entered as possible predictors of the most frequent MHP in FHCW during the first peak of the COVID-19 community transmission scenario in Mexico (Table 4), obesity, domestic violence, personal COVID-19 status, grieving because of COVID-19 and being women were included in regression models for main FHCW’s MHP during the COVID-19 peak all of them were included as predictors of all MHP. In addition, domestic violence was the main risk factor for depression followed by grieving for the death of close persons due to COVID-19 (both of which doubled the probability of its occurrence), while personal COVID-19 status constituted the principal risk factor for health anxiety/somatization.

## 4. Discussion

According to our results, and in line with previous reports in several countries, the most frequent MHP among healthcare workers dealing with COVID-19, particularly those on the frontline and women, are insomnia, depression, posttraumatic stress, and health anxiety/somatization [24,25,26,27,28,29]. These MHP have a significant negative impact on wellbeing and functionality, and are often comorbid, which could synergize their related disability. For example, in a Greek healthcare workers’ sample at the beginning of COVID-19, 73.3% of those who qualified for post-traumatic stress disorder (PTSD) according to a brief screening measure also scored positively for insomnia (as opposed to 28% of non-PTSD participants), while nearly half (48.4%) of those who met the criteria for probable PTSD also reported moderate to moderately severe depressive symptomatology (as opposed to 11% of non-PTSD sufferers) [30]. In China, as another example, Chew et al. [31] found a significant association between the prevalence of physical symptoms and anxiety, depressive, and PTSD symptoms among a wide range of healthcare workers (including doctors, nurses, associated healthcare workers, administrators, and maintenance workers). Clearly, monitoring the presence of these MHP and identifying the country-level risk factors involved in health personnel in the face of a health emergency of this nature is a prerequisite for implementing useful strategies to safeguard their mental health, which, in turn, can influence the success of medical care [32], emergency control, and social recovery [33].

In Mexico, from the outset of the pandemic [3] to the first peak of COVID-19 community transmission (analyzed in the present study), the specific frequencies of these MHP are consistent with those reported in similar studies in other countries, which also compared large samples of frontline and non-frontline healthcare workers and used screening measures to identify MHP—such as 34% of insomnia and 50.4% of depression in Chinese healthcare workers [34]. Moreover, the frequencies of all these MHP are much higher than among the national, regional, and international general populations prior to the COVID-19 pandemic [35,36,37]. As hypothesized, the frequency of depression and health anxiety/somatization among FHCW during the community transmission scenario was higher than that reported by FHCW during the clusters-of-cases scenario. This may be a result of sustained or chronic exposure to stressful events, and related to the increased experiences of mourning loved ones due to COVID-19 that are related to depressive symptoms [10] and to personal COVID-19 status, which could explain the exacerbation of health anxiety/somatization [11].

The association between these COVID-19-related variables and the presence of MHP was at least partly confirmed by the regression analysis undertaken for each of the most frequent MHP among FHCW during the community transmission peak in Mexico. Moreover, domestic violence appeared as the main risk factor for depression (doubling or even tripling the probability of its occurrence), while obesity conferred a similar or higher probability of presenting with insomnia, depression, and health anxiety/somatization than other well-known risk factors in the general population—such as being a woman [4,5].

Being a woman and having lower educational attainment (‹graduate) were other significant personal risk factors for MHP in our sample, while being ≥40 years and medical specialist or nurse (in contrast with general practitioner or medical resident) constitute protective factors. This was also consistent with previous reports on samples of healthcare workers during COVID-19 [24,25,26,27,28,29,30,38,39]. For example, Blekas et al. [30] reported that women healthcare workers scored higher on screening measures of insomnia, depression, and PTS than men; and according to several authors [40,41,42], lower age and lack of clinical experience may contribute to the development of anxiety and depression among healthcare workers dealing with COVID-19.

The limitations of this study include its cross-sectional nature, which restricts causal explanations; the potential selection bias of respondents who may experience more symptoms than non-respondents (given that the evaluation was promoted as a tool for identifying mental disorders and referring subjects to mental health treatment if needed); using self-reported screening measures; and the non-randomized neither paired sample. Although some of the main risk factors for MHP were similar in our sample of health care workers during the COVID-19 community transmission peak in Mexico and the sample of healthcare workers during the COVID-19 cluster of cases scenario (such as being female and/or single, and experiencing domestic violence), other variables that could increase at least some MHP proved higher in our sample (such as being younger and/or ‹graduate). Accordingly, the MHP frequencies encountered should not be interpreted as the prevalence of mental disorders; and the present results should be generalized with caution. Further longitudinal studies based on randomized or paired samples are required to confirm our findings (such as factors related to the increase in depression and health anxiety during the community transmission peak among FHCW).

## 5. Conclusions

Our results suggest FHCW should be closely monitored as a high-risk group for MHP and given proper prevention and treatment measures. As recommended for FHCW in many countries [11,26,43,44], preventive strategies must include measures to reduce COVID-19 contagion among FHCW (such as provision of personal protective equipment (PPE) and training in its proper use, and now that it is possible, prioritizing the vaccination of all FHCW). Inadequate PPE and the risk of contracting the disease during the pandemic have obviously subjected healthcare workers to severe psychological pressure leading to mental disorders; while the provision of PPE and effective measures to address CMD have been related to a substantial reduction in MHP among FHCW [45].

Fortunately, psychological interventions for the main MHP reported by COVID-19 FHCW have proved effective in low- and middle-income countries such as Mexico [46], and also when they are implemented during humanitarian crises [47], remotely (through videoconference) [48], and by community health workers or peers in primary care and community settings [49]. The availability of such evidence-based treatment for mood, stress-related and health anxiety/somatization for FHCW must be maintained at least during the rest of the community transmission scenario in Mexico. Additionally, professional accompaniment during experiences of grieving for the death of a close/loved one due to COVID-19 could reduce both depressive symptoms and the probability of prolonged grief disorder) [10]. Moreover, in Mexico and similar countries with a high prevalence of obesity and/or domestic violence, national policies and programs should be implemented to prevent and reduce these other health and social epidemics, which clearly affect not only the physical [1] but also the mental health status of FHCW, particularly when coping with this type of health emergencies.

## Figures and Tables

**Table 1 ijerph-19-00421-t001:** Sample characteristics.

Variable		Community Transmission Scenario (*n* = 4670)	Clusters-of-Cases Scenario(Robles et al., 2020) (*n* = 5938)
Personal and social characteristics			
Gender, *n* (%), female		3475 (74.4)	4420 (74.4)
Age, years, mean ± SD		36.6 ± 10.1 **	39.6 ± 11.9
Marital status, *n* (%)	Single	2455 (52.6)	3217 (54.2)
	Partnered	2215 (47.4)	2721 (45.8)
Education, *n* (%)	<Graduate	2917 (62.5) **	3521 (59.3)
	Graduate degrees	1753 (37.5)	2417 (40.7)
Location, *n* (%)	Metropolitan area _a_	2170 (46.5) **	2509 (42.3)
	Other states	2500 (53.5)	3429 (57.7)
Occupation, *n* (%) _b_	Nurse	2204 (47.2) **	1184 (19.9)
	GP/MR _c_	1460 (31.3) **	944 (15.9)
	Medical specialist	1006 (21.5) **	1050 (17.7)
Obesity, *n* (%)		1194 (25.6) *	1393 (23.5)
Domestic violence, *n* (%)		374 (8.0)	517 (8.7)
*COVID-19 related variables*			
Personal COVID-19 status, *n* (%)	No symptoms	4203 (90.0)	5703 (96.0)
	Suspected/confirmed COVID-19 diagnosis	467 (10.0) **	235 (4.0)
COVID-19 status among friends or relatives, *n* (%)	No symptoms	4184 (89.6)	5552 (93.5)
	Suspected/confirmed COVID-19 diagnosis	486 (10.4) **	386 (6.5)
COVID-19 Frontline HCW, *n* (%)		2218 (47.5) **	1389 (23.4)
Grieving for the death of close ones due to COVID-19, *n* (%)		981 (21.0) **	392 (6.6)

_a_ The highest number of COVID-19 cases in this phase of the pandemic in Mexico; _b_ Comparisons between each category and all other categories as a single one; summatory of percentages in Clusters-of-cases scenario (Robles et al., 2020) is not the total sample given other occupation categories were included in that study; _c_ GP: general practitioner; MR: Medical resident ** *p* ≤ 0.0001; Chi square tests (df = 2) with Yates correction or independent samples t-Student (for age) * *p* ≤ 0.05; Chi square tests (df = 2) with Yates correction.

**Table 2 ijerph-19-00421-t002:** Mental health problems.

Mental Health Problems	Community Transmission Peak			Clusters-of-Cases Scenario
	Total Sample *n* = 4670	Frontline HCW		Frontline HCW (Robles et al., 2020) *n* =1389
		No *n =* 2452	Yes *n* = 2218	
Insomnia, *n* (%)	2136 (45.7)	973 (39.8)	1163 (52.4) ***	723 (52.1)
Depression, *n* (%)	1747 (37.4)	783 (32.0)	964 (43.4) ***	524 (37.7) **
Posttraumatic stress, *n* (%)	1582 (33.9)	689 (28.2)	893 (40.3) ***	521 (37.5)
Health anxiety and somatization, *n* (%)	993 (21.3)	414 (16.9)	579 (26.1) ***	306 (22) **
Generalized anxiety, *n* (%)	604 (12.9)	262 (10.7)	342 (15.4) ***	219 (15.8)

*** *p* ≤ 0.0001; Chi square tests between frontline and non-frontline in community transmission scenario ** *p* ≤ 0.01 Chi square tests between frontline in community transmission scenario and frontline in clusters-of-cases scenario.

**Table 3 ijerph-19-00421-t003:** FHCW during COVID-19 peak: Bivariate comparisons by main MHP _a_ (*n* = 2218).

Variable and Categories		Insomnia		Depression		Posttraumatic Stress Disorder		Health Anxiety Somatization	
		Yes	No	Yes	No	Yes	No	Yes	No
		*n =* 1163	*n =* 1055	*n =* 964	*n =* 1253	*n =* 1633	*n =* 583	*n* = 1633	*n* = 583
		%	%	%	%	%	%	%	%
Gender	Women	78.2 ***	68.8	78.1 ***	70.3	81.0 ***	68.8	78.2 **	72.1
	Men	21.8	31.2	21.9	29.7	19.0	31.2	21.8	27.9
Age	≤39 years	73.0 **	67.3	73.9 **	67.5	74.2 **	67.7	70.1	70.4
	≥40 years	47.6	52.4	26.1	32.5	25.8	32.3	29.6	29.9
Marital status	Single	55.5 *	51.0	56.1*	51.2	56.1 *	51.5	52.2	53.8
	Partnered	44.5	49.0	43.9	48.8	43.9	48.5	47.8	46.2
Education	<Graduate	57.1 ***	64.4	58.2 *	62.3	56.6 **	63.2	62.0	60.0
	≥Graduate	42.9	35.6	41.8	37.7	43.4	36.8	40.0	38.0
Occupation 1	Nurse	47.5	52.8	45.3	53.6	47.9	51.4	51.5	49.5
	Other (GP/MR _b_ or MS _c_)	52.5 *	47.2	54.7 ***	46.6	52.1	48.6	48.5	50.5
Occupation 2	GP/FD/MR_b_	25.3	19.5	24.1	21.4	24.6	21.2	22.3	22.7
	Other (nurse or MS _c_)	74.7 **	80.5	75.9	76.6	75.4	78.8	77.7	77.3
Occupation 3	Medical specialist	47.5	52.8	45.3 ***	53.6	47.9	51.4	51.5	49.5
	Other (nurse or GP/MR _b_)	52.5 **	47.2	54.7	46.4	52.1	48.6	48.5	50.5
Obesity	Yes	30.4 ***	20.7	30.8 ***	21.9	28.2 **	24.0	35.5 ***	23.0
	No	69.6	79.3	69.2	78.1	71.8	76.0	66.5	77.0
Domestic violence	Yes	10.7 ***	5.9	12.1 ***	5.6	10.7 **	6.9	12.3 ***	7.1
	No	89.3	94.1	87.9	94.4	89.3	93.1	87.7	92.9
COVID-19 personal status	Yes	16.2 ***	9.8	16.6 ***	10.4	16.1 **	11.1	23.0 ***	9.7
	No	83.8	90.2	83.4	89.6	83.9	88.9	77.0	90.3
COVID-19 in close persons	Yes	16.1 ***	10.7	15.2 *	12.2	15.6 *	12.2	16.6 **	12.5
	No	83.9	89.3	84.8	87.8	84.4	87.8	83.4	87.5
Grieving due to COVID-19	Yes	29.5 ***	17.9	31.2 ***	18.4	27.8 ***	21.3	34.9 ***	20.2
	No	70.5	82.1	68.8	81.6	72.2	78.7	65.1	79.8

_a_ Results for variables with that showed differences by category in at least one MHP; _b_ GP: general practitioner; MR: Medical resident; _c_ MS: medical specialist *** *p* ≤ 0.0001; Chi square tests (df = 1) ** *p* ≤ 0.01; Chi square tests (df = 1) * *p* ≤ 0.05; Chi square tests (df = 1).

**Table 4 ijerph-19-00421-t004:** Regression models to predict most frequent CMD in Mexican FHCWs during COVID-19 peak (*n* = 2218).

	Insomnia		Depression		Posttraumatic Stress		Health Anxiety Somatization	
	Unconditional OR (95%CI)	Final Model OR (95%CI)	Unconditional OR (95%CI)	Final Model OR (95%CI)	Unconditional OR (95%CI)	Final Model OR (95%CI)	Unconditional OR (95%CI)	Final Model OR (95%CI)
Gender, women	1.277 ** (1.079–1.512)	1.757 *** (1.432–2.155)	1.063 (0.911–1.241)	1.692 *** (1.373–2.084)	0.873 * (0.766–0.995)	2.025 *** (1.643–2.496)	0.335 *** (0.292–0.385)	1.385 ** (1.096–1.751)
Age, ≥40 years	0.585 *** (0.482–0.709)	0.621 *** (0.510–0.756)	0.538 *** (0.445–0.651)	0.623*** (0.511–0.758)	0.541*** (0.446–0.655)	0.623 *** (0.511–0.760)		
Education, <graduate	1.04 (0.879–1.245)	1.361 ** (1.116–1.660)			1.020 (0.864–1.203)	1.471 *** (1.227–1.765)		
Medical specialist	0.649 *** (0.544–0.773)	0.818 * (0.673–0.994)						
Nurse			0.538 *** (0.453–0.637)	0.639 *** (0.534–0.765)				
Obesity	1.597 *** (1.300–1.939)	1.769 *** (1.442–2.170)	1.420 *** (1.167–1.728)	1.639 *** (1.341–2.004)	1.028 (0.848–1.246)	1.296 ** (1.061–1.582)	0.989 (0.807–1.210)	1.649 *** (1.329–2.045)
Domestic violence	1.584 ** (1.217–2062)	1.720 ** (1.238–2.389)	1.878 *** (1.365–2.584)	2.093 *** (1.519–2.885)	1.275 (0.941–1.727)	1.512 ** (1.110–2.059)	1.200 (0.871–1.652)	1.628 ** (1.174–2.258)
COVID-19 personal	1.584 ** (1.217–2.062)	1.760 *** (1.346–2.301)	1.388 ** (1.076–1.792)	1.578** (1.219–2.043)	1.189 (0.927–1.524)	1.439 ** (1.115–1.856)	1.768 *** (1.365–2.290)	2.819 *** (2.162–3.677)
COVID-19 close persons	1.452 ** (1.123–1.878)	1.659 *** (1.275–2.158)					0.955 (0.730–1.250)	1.563 ** (1.183–2.065)
Grieving due to COVID-19	1.625 *** (1.318–2.002)	1.768 *** (1.430–2.185)	1.684 *** (1.373–2.062)	1.945*** (1.582–2.391)	1.121 (0.918–1.368)	1.339 ** (1.091–1.643)	1.191 (0.967–1.467)	1.823 *** (1.466–2.266)
Constant		0.523		0.496		0.313		0.156
R2 Nagelkerke		0.100 (second step)		0.097 (first step)		0.063 (first step)		0.093 (first step)
Hosmer-Lemeshow (*p* value)		0.300 second step)		0.164 (first step)		0.065 first step)		0.511 (first step)

*** *p* ≤ 0.0001 ** *p* ≤ 0.01 * *p* ≤ 0.05.

## Data Availability

The data presented in this study are available on request from the corresponding author.

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
