# Peer review of "Mental Health Problems among COVID-19 Frontline Healthcare Workers and the Other Country-Level Epidemics: The Case of Mexico"

_ijerph, 2021, doi:10.3390/ijerph19010421_

Round 1

Reviewer 1 Report

The authors have done a fine job in responding to reviewers' queries and suggestions. Other than that, I have no further comments

Author Response

Thank you very much for your kind and expert evaluation. 

Reviewer 2 Report

Dear Authors,

Thank you for sharing a revised version of the manuscript. The revised manuscript reflects my comments and recommendations. I suggest two additional revisions.

  • The results section reports the Cronbach’s alpha of the measures. It is recommended to also mention in the methods section that Cronbach's alpha was used to evaluate the internal consistency of each of the used scales.
  • The revised Table 4 present the crude and adjusted ORs and 95%CI of Insomnia in two columns. It is strongly recommended to also show in different columns the crude and adjusted results of Depression, Posttraumatic Stress, and Health anxiety somatization. The current Table suggests that unconditional and final ORs and 95%CIs were the same which is very unlikely. In addition, Table 4 is not mentioned in the text of the results section. This table could be cited in the last paragraph of the results section.

Thank you.

Author Response

Comment 1. The results section reports the Cronbach’s alpha of the measures. It is recommended to also mention in the methods section that Cronbach's alpha was used to evaluate the internal consistency of each of the used scales.

Response:  

Thank you very much for this suggestion. In the new version of the manuscript, specifically in the second paragraph of the subsection for procedures and data analyses in methods section, we added the information about the use of Cronbach´s alpha coefficients to analyze the internal consistency of the measures in our sample in the methods section (in track of changes), as follows (in red):    

All analyses were performed using SPSS Statistics 21.0. First, we obtained descriptive statistics to characterize the sample, using means and standard deviations to describe continuous variables and frequencies and percentages to summarize categorical ones. Cronbach´s alpha coefficients were used to analyze the internal consistency of the measures in our sample. Independent samples, Student-t tests, and chi square tests were used to compare these variables between the present sample of healthcare workers during the first peak of the community transmission scenario in Mexico, and Robles et al.’s [3] sample of healthcare workers during the clusters of cases scenario.

Comment 2: The revised Table 4 present the crude and adjusted ORs and 95%CI of Insomnia in two columns. It is strongly recommended to also show in different columns the crude and adjusted results of Depression, Posttraumatic Stress, and Health anxiety somatization. The current Table suggests that unconditional and final ORs and 95%CIs were the same which is very unlikely. In addition, Table 4 is not mentioned in the text of the results section. This table could be cited in the last paragraph of the results section.

Response: Thank you for noticing this involuntary mistake. The new table 4 includes different columns showing the crude and adjusted results for all the conditions. In addition, Table 4 is referred in the last paragraph of the results section (in track of change) as follows (in red):

Moreover, when these personal, social and COVID-19-related variables were entered as possible predictors of the most frequent MHP in FHCW during the first peak of the COVID-19 community transmission scenario in Mexico (table 4), obesity, domestic violence, personal COVID-19 status, grieving because of COVID-19 and being women were included in regression models for main FHCW´s MHP during the COVID-19 peak all of them were included as predictors of all MHP. In addition, domestic violence was the main risk factor for depression followed by grieving for the death of close persons due to COVID-19 (both of which doubled the probability of its occurrence), while personal COVID-19 status constituted the principal risk factor for health anxiety/somatization.

This manuscript is a resubmission of an earlier submission. The following is a list of the peer review reports and author responses from that submission.

Round 1

Reviewer 1 Report

The study examined a number of mental health conditions among frontline healthcare workers in Mexico during the COVID-19 pandemic. 

Data were from a convenience sample recruited online between 8th May to 18th August 2021. The authors acknowledge a number of important limitations, not least the biased sample and the fact that the study is cross-sectional. As such, the study is very context-specific and not generalisable beyond that context (again acknowledged by the authors). All that said, the findings do show the extent of mental health problems among frontline healthcare workers in Mexico and is important in that regard. 

Overall, the manuscript is well written and is a useful descriptive piece of analysis on an important topic.

L170 & 292: The regressions are multivariable, not multivariate. The latter refers to models with more than one outcome whereas each of your regressions has one outcome at a time and more than one independent variables i.e. multivariable.

Table 3. Please replace column percentages with row %s. Take gender for example, the important thing to report here is not the % of men wna women within each of the yes/no categories, but instead the percentage of yes/no among men and women. That is what shows if there is a gender difference or not.

Some typos:
L36: In concussion
L51: being obesity
L71: closed ones

Author Response

Point 1: The regressions are multivariable, not multivariate. The latter refers to models with more than one outcome whereas each of your regressions has one outcome at a time and more than one independent variables i.e. multivariable.

Response: Thank you very much for this clarification. The word has been changed. 

Point 2: Table 3. Please replace column percentages with row %s. Take gender for example, the important thing to report here is not the % of men wna women within each of the yes/no categories, but instead the percentage of yes/no among men and women. That is what shows if there is a gender difference or not.

Response: Table 3 has been corrected. We appreciate this suggestion, now the table is more clear about differences between the categories of each variable. 

Point 3: Some typos:
L36: In concussion
L51: being obesity
L71: closed ones

Response: Thank you, all typos has been corrected. 

Reviewer 2 Report

The manuscript entitled “Mental health problems among COVID-19 frontline healthcare workers and the other country-level epidemics: The case or Mexico” aimed to determine the frequency of insomnia, depression, posttraumatic  stress  symptoms, health anxiety/somatization and generalized anxiety symptoms among frontline healthcare  workers  (FHCPs). The stated specific study objectives included “1) comparing the frequency of MHP between FHCW and healthcare workers not working directly with COVID-19 patients during the peak of the community transmission  scenario  in  Mexico;  2)  comparing  FHCW’s  frequency  of  MHP  between  the COVID-19 clusters of cases scenario (April 17 to May 07 2020) and the peak of COVID-19 community transmission scenario in Mexico  (May 08  to  August 18 2020); and  3) determining the level of risk conferred by personal, social and COVID-19-related variables on the main MHP reported by FHCW during the peak of the COVID-19 community transmission scenario in Mexico”. A cross sectional survey was designed with Mexican healthcare workers and completed between May 8 and August 18, 2020. Mexican HCWs were invited to participate in the survey through official media, press conferences, and the National Institute of Psychiatry’s website and social networks.

I consider this work valuable; although, the current version of the manuscript has major issues that need to be revised before considering this work for publication. 

Major issues

  1. I recommend revising the aim and specific objectives of the study. For clarity, the wording of the objectives could be summarized. This change will guide better the readers.
  2. Details of the study target population, specifically Mexican healthcare workers, are needed. The study sample included nurses, general practitioners, family doctors, medical residents, medical specialists, administrative staff, psychologists, psychology students, paramedics, social workers, cleaning and laundry staff, food preparation staff, laboratory and blood bank staff. In addition, the “other” category included “gurney operator, rescue, hospice and morgue staff”. The manuscript must include details of the total number of healthcare workers in Mexico for each of these categories.
  3. Study design and procedures need to be revised and expanded.
    1. I suggest moving procedures content (i.e., lines 140 to 150) to the first paragraph of the materials and methods section. Information about study procedures should be presented before talking about the study variable and measures.
    2. An online survey was used. It is strongly recommended to provide more details of the platform that was used for designing the online survey and collecting the data.
    3. The manuscript must state if participation in the online survey was anonymous or anonymized, as well as how duplicated responses were controlled.
    4. There are no details about missing data and how missing data were handled in the study.
    5. Please provide further details of the clusters of cases scenario in the methods section. A reference is cited; although, the readers will appreciate details of this approach is explained in more detail within this manuscript.
  4. Despite previously developed and validated questionnaires were used, I strongly recommend evaluating and reporting the reliability of each of the scales within the study sample. This adjustment should be reflected in the methods and results sections of the manuscript.
  5. A subsample was used in the multivariable analysis, as evidence in Table 3 (n=2,976). It appears this sample includes frontline healthcare workers only. Please describe in the methods section the selection criteria f this subsample.
  6. It was acknowledged the insufficient sample size of certain occupational categories (i.e., paramedics, gurney operators, cleaning, laundry, food preparation, laboratory and blood bank staff). The sample descriptive statistics could include these groups. Notwithstanding, it is strongly recommended to exclude the data of these occupations (and likely from other underrepresented groups depending on numbers requested in point #2 above) from the other study analyses. Subsequently, analyses should be run with the adjusted sample and revised results must be presented. This approach could reduce the biases of underrepresented groups in the study sample.
  7. It is recommended to also present unconditional ORs and corresponding 95%CI. I suggest doing this in a new or existing table.
  8. In the methods, it is described how the model’s goodness of fit was evaluated. Although, these results are not discussed.
  9. It is stated in the methods that a convenience sample was used. Researchers also described that Mexican HCWs were invited to participate in the survey through official media, press conferences, and the National Institute of Psychiatry’s website and social networks. The limitations of using a convince sample and recruitment method need to be expanded and discussed.
  10. It is recommended to include as a supplementary file a copy of the full questionnaire used in the study. Readers will appreciate the opportunity to see the tool used in this study.

Minor issues

  1. There might be a typo in the title of the manuscript. The title currently says, “The case or Mexico”. I assume the authors meant “The case of Mexico”.
  2. All tables must state the sample size included for the corresponding set of results presented, similar to Table 3.

Author Response

Point 1: I recommend revising the aim and specific objectives of the study. For clarity, the wording of the objectives could be summarized. This change will guide better the readers.

Response: We appreciate this suggestion. The aim and objectives have been sumarized as follows: ... the aim of this study was to determine the frequency of MHP and related personal, social and COVID-19-related risk factors among FHCW during the first peak of the COVID-19 community transmission scenario in Mexico. Specific objectives included: 1) comparing the frequency of MHP between FHCW and healthcare workers not working directly with COVID-19 patients; 2) comparing FHCW’s frequency of MHP between the COVID-19 clusters of cases scenario and the peak of COVID-19 community transmission scenario; and 3) determining the level of risk conferred by personal, social and COVID-19-related variables on the main MHP reported by FHCW during the peak of the COVID-19 community transmission scenario in Mexico.

Point 2:  Details of the study target population, specifically Mexican healthcare workers, are needed. The study sample included nurses, general practitioners, family doctors, medical residents, medical specialists, administrative staff, psychologists, psychology students, paramedics, social workers, cleaning and laundry staff, food preparation staff, laboratory and blood bank staff. In addition, the “other” category included “gurney operator, rescue, hospice and morgue staff”. The manuscript must include details of the total number of healthcare workers in Mexico for each of these categories.

Response: According to the  reviewer´s recommended to exclude the data of  underrepresented groups (point 6), the new version of the manuscript includes only nurses, general practitioneres, medical residents and medical specialists. At the end of the third paragraph of the Materials and Methods section, this target population has been described as follows:  In 2020, according to the Mexican Ministry of Health, the study target population includes, 286,370 nurses, 270,600 general practitioners, 22,613 medical residents, and 147,910 medical specialists.

Point 3a: I suggest moving procedures content (i.e., lines 140 to 150) to the first paragraph of the materials and methods section. Information about study procedures should be presented before talking about the study variable and measures.

Response: In the new version of the manuscript, such paragraph is the first of the materials and methods section. 

Point 3b: An online survey was used. It is strongly recommended to provide more details of the platform that was used for designing the online survey and collecting the data.

Response: In the new version of the manuscript a more datailed desciption of the plattaform is included (at the end of the first paragraph of Materials and Methods section): ... programmed on a platform (https://www.misalud.unam.mx/covid19/) developed in Linux®, PHP®, HTML®, CSS®, and JavaScript®), which encrypts the information under the computer security standards of the Mexican laws on protection of personal data.

Point 3c: The manuscript must state if participation in the online survey was anonymous or anonymized, as well as how duplicated responses were controlled.

Response: We clarified these important issues in the last phrase of the first paragraph of materials and methods section as follows: (after explaining that the platforms enclypt personal information)... This allowed to ask for participants Unique Population Registry Code when log in in the system to ensure traceability of participation although it was anonymized.

Point 3d: There are no details about missing data and how missing data were handled in the study.

Response: We clarified (in the second phrase of the second paragraph of Methods and materials section) that: During the survey, participants must answer all items to be allowed to proceed to the next question (to avoid missing data). 

Point 3d: Please provide further details of the clusters of cases scenario in the methods section. A reference is cited; although, the readers will appreciate details of this approach is explained in more detail within this manuscript.

Response: Clusters of cases scenario has been definen (according to WHO) in the second paragraph of the introduction given that the studies carried out in this stage was the antecedents to this study, which was done in the community transmission scenario (describen in general at the introduction and in more detail in the methods and materials subsection). 

Point 4: Despite previously developed and validated questionnaires were used, I strongly recommend evaluating and reporting the reliability of each of the scales within the study sample. This adjustment should be reflected in the methods and results sections of the manuscript.

Response: Thnak you for this valuable suggestion. We included the information of the analyses and the results in both sections. At the end of the second paragraph of the data analyses subsection we clarified the method to do so as follows: ... internal consistencies as a psychometric measure of reliability was calculated for all the instruments employed in our specific sample using Cronbach´s alpha coefficients. At the end of the second paragraph of the Results section, we descibed our findings in this regard as follows: According to our analyses on the internal consistency of the measures employed to evaluate the presence of these mental disorders showed high Cronbach alphas in our sample, being 0.939 for PCL-5, 0.849 for the Anxiety Scale for ICD-11 PHC field studies, 0.897 for the first eight items in the SSOM Current Status Assessment Questionnaire, and 0.874 for PCL-2.

Point 5:  A subsample was used in the multivariable analysis, as evidence in Table 3 (n=2,976). It appears this sample includes frontline healthcare workers only. Please describe in the methods section the selection criteria f this subsample.

Response: Yes, it is the subsample of frontline healthcare workers. In the variavles and measures subsetion of Materials and Methos, the criteria to indicate that the participant is a COVID-19 frontline healthcare worker has been included ((i.e. working directly treating patients with COVID-19).

Point 6: It was acknowledged the insufficient sample size of certain occupational categories (i.e., paramedics, gurney operators, cleaning, laundry, food preparation, laboratory and blood bank staff). The sample descriptive statistics could include these groups. Notwithstanding, it is strongly recommended to exclude the data of these occupations (and likely from other underrepresented groups depending on numbers requested in point #2 above) from the other study analyses. Subsequently, analyses should be run with the adjusted sample and revised results must be presented. This approach could reduce the biases of underrepresented groups in the study sample.

Response: We appreciate this recomendation. We have run all the analyses using a new sample conformed only by nurses, general practitioners, medical residents and medical specialists. Revised results are presented (highligted in red) along the manuscript  (highligted in red), and corresponding changes has been done in all relevant sections (also highlighted in red). As can be seen, general results do not change, but now we have confirmed and presented them reducing the possible biases of a sample including underrepresented groups (as reviewer stated).

Point 7: It is recommended to also present unconditional ORs and corresponding 95%CI. I suggest doing this in a new or existing table.

Response: In the new version of the manuscript, unconditional ORs and corresponding 95% CI are included in table 4.

Point 8: In the methods, it is described how the model’s goodness of fit was evaluated. Although, these results are not discussed.

Response: In the new version of the manuscript we discussed the results of the model´s goodness of fit in the fourth paragraph of the discussion section as follows (highlighted in red): ... giving final models with an adequate predictive ability (according Hosmer-Lemeshow test). 

Point 9: It is stated in the methods that a convenience sample was used. Researchers also described that Mexican HCWs were invited to participate in the survey through official media, press conferences, and the National Institute of Psychiatry’s website and social networks. The limitations of using a convince sample and recruitment method need to be expanded and discussed.

Response: The limitations of using a convience sample and recruitment method have been recognized and discussed as follows (in last paragraph of discussion section dedicated to limitations, higlighted in red): ... the use of a convenience sample, which prevents the generalization of the results to the population as a whole; and potential selection biases given participants´ recruitment through media, websites and social networks and based on the promotion of a tool for identifying MHP and referring those in need to mental health treatment. Restricted access to telecommunication services in distant rural settings (which might decrease healthcare workers living or working there opportunity to be informed about the services offered), as well as higher participation from those who may experience more mental health symptoms cannot be ruled out. Particularly, MHP frequencies encountered should not be interpreted as the prevalence of mental disorders... 

Point 10: It is recommended to include as a supplementary file a copy of the full questionnaire used in the study. Readers will appreciate the opportunity to see the tool used in this study.

Response: Questionnaire is still available at the official COVID-19 website (coronavirus.gob.mx/salud-mental/), which is refered in the text (first paragraph of Materials and Methods section). Additionally, all measures employed are available in English in the corresponding references included in the manuscript. 

Round 2

Reviewer 1 Report

Thank you for addressing all of my previous comments with the exception of the row/column % issue in Table 3.

Apart from again suggesting that this is addressed I have no further comments.

Reviewer 2 Report

Dear Authors,

Thank you for addressing the provided comments and recommendations. As follow up recommendations, please consider to:

A.    Include a reference that supports the statement in line 110
B.    Revise information of the used platform for the online survey, its anonymized nature, and how missing data was managed in the Methods section. It appears that the adjustments were not included in the revised manuscript (see points 3b, 3c, and 3d, as well as the authors’ responses)
C.    Revise details of the reliability of the questionaries in the Methods and Results sections. The authors’ response addressed the comments, but the revised version of the manuscript did not include the revisions (see point 4 and authors’ comments)
D.    Revise Table 4; the revised table appears to be reporting adjusted ORs only (see point 7 and authors’ comments)
E.    Include a PDF copy of the study questionnaire (point 10). The questionnaire might be available online; although, it could not be seen unless an individual completes the form.

Thank you.